# Translational Roadmap for the Organs-on-a-Chip Industry toward Broad Adoption

**DOI:** 10.3390/bioengineering7030112

**Published:** 2020-09-16

**Authors:** Vanessa Allwardt, Alexander J. Ainscough, Priyalakshmi Viswanathan, Stacy D. Sherrod, John A. McLean, Malcolm Haddrick, Virginia Pensabene

**Affiliations:** 1Center for Innovative Technology, Department of Chemistry, Vanderbilt University, Nashville, TN 37212, USA; vanessa.allwardt@Vanderbilt.Edu (V.A.); stacy.d.sherrod@vanderbilt.edu (S.D.S.); john.a.mclean@Vanderbilt.Edu (J.A.M.); 2National Heart and Lung Institute, Imperial College London, London SW3 6LY, UK; alex.ainscough14@imperial.ac.uk; 3Medicines Discovery Catapult, Alderley Park, Alderley Edge, Macclesfield SK10 4TG, UK; priya.viswanathan@md.catapult.org.uk (P.V.); malcolm.haddrick@md.catapult.org.uk (M.H.); 4Vanderbilt Institute of Chemical Biology, Vanderbilt-Ingram Cancer Center, Vanderbilt Institute for Integrative Biosystems Research and Education, Vanderbilt University, Nashville, TN 37235, USA; 5School of Electronic and Electrical Engineering, School of Medicine, Leeds Institute of Medical Research at St. James’s, University of Leeds, Leeds LS2 9JT, UK

**Keywords:** organ-on-a-chip, technology-led strategy, gap analysis, validation, translation, quality management, body-on-a-chip, organotypic culture models, microphysiological systems

## Abstract

Organs-on-a-Chip (OOAC) is a disruptive technology with widely recognized potential to change the efficiency, effectiveness, and costs of the drug discovery process; to advance insights into human biology; to enable clinical research where human trials are not feasible. However, further development is needed for the successful adoption and acceptance of this technology. Areas for improvement include technological maturity, more robust validation of translational and predictive in vivo-like biology, and requirements of tighter quality standards for commercial viability. In this review, we reported on the consensus around existing challenges and necessary performance benchmarks that are required toward the broader adoption of OOACs in the next five years, and we defined a potential roadmap for future translational development of OOAC technology. We provided a clear snapshot of the current developmental stage of OOAC commercialization, including existing platforms, ancillary technologies, and tools required for the use of OOAC devices, and analyze their technology readiness levels. Using data gathered from OOAC developers and end-users, we identified prevalent challenges faced by the community, strategic trends and requirements driving OOAC technology development, and existing technological bottlenecks that could be outsourced or leveraged by active collaborations with academia.

## 1. Introduction

The organ-on-a-chip (OOAC) concept emerged about 10 years ago when scientists combined fluidic systems, cell culture techniques, analytical methods, and single, 2D, and 3D cell culture protocols into new in vitro models. OOAC systems were developed to recapitulate typical functions of human organs in microliter volumes. The potential of these models, also defined as “microphysiological systems”, attracted research groups and pharmaceutical companies looking for more effective, efficient, and cost-saving techniques to reduce drug development failures. Subsequently, a multitude of different microfluidic systems have been designed and successfully developed, as depicted in Figure 1. While some examples have already advanced our understanding of human physiology, pharmacology, and toxicology, a large percentage of OOAC models are still far from optimization, and their application is only guaranteed in very limited and specific working conditions.

The OOAC market has been recently valued at 21 Million US$ and is projected to reach 220 Million US$ by 2025 [3]. Most of the companies (start-ups and large companies) reside in the United States, UK, The Netherlands, and France, while new research is sprouting in South Korea, Japan, and Taiwan. The highest development has been seen in heart-on-chip, human-on-chip, intestine-on-chip, kidney-on-chip, liver-on-chip, and lungs-on-chip [4].

The technology development path is commonly non-linear in academic research, and naturally, many projects stop at the proof of concept stage. The effective industrial development of a product needs, instead, to steadily proceed and be completed in the shortest possible time before the public and commercial interests deflate and new solutions appear on the market. A key factor to a linear technology development pathway is to define the value proposition, to promptly transition from an R&D-only stage to the validation phase, and finally move to optimization and scale-up for rapid adoption. These steps are necessary to deliver the promised benefits and to successfully exploit funding sources and investments [5]. Mistakes and delays along this path lead to continuous technological challenges and revisions of the business model, but most importantly, risk attenuating the interest of end-users and reduce future funding opportunities.

Using the organs-on-a-chip technologies network, established in the United Kingdom in 2018 as an indicative collection of experts in the field of OOAC, and the 3DbioNet network, started in 2019 in the United Kingdom under the technology touching life initiative, we surveyed end-users, experts, and developers from both academia and industry about the existing stage of development, the faced challenges, and the limitations of organs-on-a-chip.

From the responses, we aimed to identify essential values of the OOAC technology recognized by active users and researchers in the field, to complement these with an analysis of the most developed and most promising models in the peer-reviewed literature. This review also focused on the specific technological expectations, characteristics defined as “needed” by the users, and their comparison with the products and technological challenges recognized by developers in the academic and industrial fields.

By interviewing both academics and product development specialists in the industry, this review provided an analysis of the perceived hindrances to translation and commercialization. Technological aspects, integration activities, and requirements for adoption and acceptance by the pharmaceutical industry were identified and discussed, as well as describing the potential focus of future joint collaborations between academia and industry to translate OOAC into commercial success.

## 2. Translational OOAC Company and End-User Survey

For the anonymously conducted expert survey, data were sampled and collected in two stages. First, we identified and contacted 64 companies currently developing OOAC technology, as well as over 270 end-users from the pharmaceutical industry, European and the United States regulatory agencies, the 3DbioNet network, and organ-on-a-chip technologies network in the United Kingdom. A web search analysis was used to complement the list of members of the OOAC Technologies Network (OOATC). Organizations were categorized as contract research organizations (CROs), pharmaceutical companies, life sciences companies, universities, and research institutes. For each of these organizations, contacts (names and e-mail addresses) were extracted from company websites or within the authors’ professional networks. In a second step, direct connections with the identified companies were established via LinkedIn, providing a short summary of the goal of our research. In its first form, the survey was pilot-tested by the authors of this manuscript and then distributed amongst end-users and developers from academia and industry.

Of the 524 requests that were sent, we received a total of 140 responses, providing a response rate of 26.7% that is within the typical range of research organizational analyses (0.36 ± 0.20) [6]. After data cleaning, the completed surveys were 85 (completion rate 16.2%): 50 respondents self-identified as OOAC-centric developers, 28 respondents as end-users, and seven respondents as developers of OOAC-related technology as part of a broader product portfolio (full data in Appendix A).

The 27 questions were diversified for developers and users and included either multiple-selection, single choice, or 5-points Likert scale answers. For non-normally distributed data, non-parametric tests were used. The mean, standard deviation, and median were calculated for questions using the 5-point Likert scale. The final scores for each answer were multiplied by its respective weight.

Statistical differences between OOAC developers’ and end-users’ responses were assessed by using a t-test or Kruskal–Wallis H-test, as appropriate. Microsoft Excel was used for the statistical analyses, and a *p*-value ≤ 0.05 was considered statistically significant. Of the OOAC developers, an equal number of participants (92% total) identified their core area of business as either “academic/non-profit” or as “OOAC company” (Figure 2). Most end-user respondents were from non-profit academic research (35.7%) and the pharmaceutical industry (32.1%) sector. Half of all surveyed developers were from companies or research groups with less than 10 employees, 34% in companies or research groups with 10–50 employees, and 12% in companies with more than 100 employees.

## 3. Organs-on-a-Chip and Their Recapitulation of Biological Functions

Aristotle postulated: the whole is greater than the sum of its parts. Overall, organs conduct a multitude of specialized and interconnected functions simultaneously with each function given the courtesy of key functional units. To model an organ using an OOAC, multiple tissues composed of different cell types are required to interact within a microenvironment [7]. These interactions can be facilitated directly by employing contact or non-contact tissue-tissue interfaces, or indirectly by connecting chambers containing individual tissues. Individual tissues on microfluidic platforms, also referred to as parenchyma-on-chip, can be studied, notably for endothelial cell sprouting in angiogenesis, which is reviewed elsewhere [8].

Organs-on-chips are used to commonly model interactions between a tissue-specific parenchymal layer and an endothelial layer, or between parenchymal tissue of two or more organs.

Figure 3 provides an overview of OOAC models currently offered and used by survey respondents and which models are of interest to end-users for future use. The most intensive research has been directed towards organs most pertinent to pharmacological adsorption, distribution, metabolism, and excretion (ADME) and cytotoxicity studies, and these include lung, gut, kidney, heart, and liver [9]. However, many research endeavors have expanded the field to model many different organs.

Of the different drug administration routes, skin-on-a-chip technologies could be an endocrine model of interest to ADME and cytotoxicity research. These models may also offer a partial solution for cosmetic companies with longstanding friction, with animal rights groups, by reducing the number of animal testing procedures [10]. Skin is the largest organ in the body by area. Common skin ailments involve inflammation, which has been simulated in a 3-layer microfluidic device using tumor necrosis factor-alpha (TNFα) and treated using dexamethasone [11].

Recently, research endeavors have successfully modeled the female reproductive system with the individual ovary, fallopian tube, uterus, cervix, endometrium, fetal membrane, and liver compartments sustained over a 28-day period [2,12,13]. Using tissue ex vivo on a novel microfluidic platform, researchers have been able to emulate a pregnancy-like progesterone secretion profile, presenting the huge potential for in vitro pharmacology studies. The maintenance of the placental barrier in fetal development is crucial; thus, the development of a placenta-on-a-chip, which recapitulates the maternal-to-fetal transfer of glucose, may provide mechanistic insights into the regulation of placental barrier function [14]. Additional models of the female reproductive system have been extensively reviewed elsewhere [15].

Muscle wasting can arise due to genetic conditions, reduced usage during aging, damage through a sports injury, or surgical interventions, among other causes. Skeletal muscle-on-chip platforms have potential in the development of novel therapies. Quantification of muscle tension has been achieved in one such platform, where cardiotoxin-induced changes in tissue architecture have been demonstrated in a dose-dependent manner [16]. Another platform with an integrated biosensor has been able to monitor levels of secreted interleukin-6 (IL-6) or TNFα with the potential for investigating mechanisms of inflammation that hinder tissue regeneration [17].

Numerous groups have developed functional OOAC models that are excellent at replicating aspects of specific diseases, aligning with the broader objective of OOAC technology: to produce biomimetic models of organs that recapitulate key organ functions. Accurate disease modeling is crucial for obtaining data reflective of in vivo physiologies [18]. Diversification of models for specific diseases represents a major challenge for both developers and end-users. One challenge lies in that device versatility and specificity often depend on the objective of the research team. A universal OOAC device design may over-simplify modeling by failing to capture the complexity or the scale of the target tissue. Device architecture, even from the same research group, varies depending on the organ of interest; for example, the model of a gut-chip may have a serpentine-like fluidic design, whereas the model of a kidney-chip may be more linear [19].

The versatility of organ-on-chip systems provides opportunities for modeling numerous diseases that may be less studied (or less common) and advancing treatments for these conditions.

### 3.1. Advanced OOAC Disease Models

Liver OOAC models have been extensively studied, owing to the inability of current preclinical in vivo animal models to accurately predict the high proportion of human drug-induced liver injury. Ehrlich et al. published a critical review of existing, commercially available liver-on-a-chip devices, evaluating their advantages and problem areas, which include excessive shear stresses, low oxygen transport due to slow media flow rates, and nonspecific chemical absorption by silicone components [20]. Yet, several groups have been able to recapitulate clinically relevant tissue responses, showing improvements over conventional 2D cultures [21,22].

Perhaps the most well-known OOAC device is the alveolus-on-a-chip (or “breathing lung-on-a-chip”) model [23], in which IL-2-inducible vascular leakage as a consequence of a compromised endothelial-epithelial barrier, mimicking pulmonary edema, was initially characterized. Pulmonary edema is commonly observed in acute respiratory distress syndrome (ARDS), for which there are currently no approved pre-clinical models; thus, OOACs offer immense potential value by enabling testing of pharmaceutical interventions to bridge this therapeutic gap. In a follow-up study, the alveolus on-a-chip, without the use of cyclic stretching, was shown to be an effective model for in vitro thrombus formation, giving it potential use as a platform to identify and validate novel antithrombotic drugs [24].

Another alveolus-on-a-chip model similarly employs an endothelial-epithelial tissue-tissue interface and presents a potentially higher throughput platform. Six independent alveolar models are cultured in parallel within a single microfluidic device, demonstrating enhanced barrier integrity after 22 days in co-culture as assessed by transepithelial/endothelial electrical resistance (TEER) and fluorescein isothiocyanate (FITC) or tetramethylrhodamine-isothiocyanate (TRITC) dextran measurements [25]. A challenge for pharmaceutical companies here is the existence of OOAC models of the same region using the same tissue types but with slightly different technologies (e.g., differing geometries, materials, scales)—which model should be adopted?

A novel 3D scaffold bioreactor platform has reported a stable co-culture of hepatocytes and Kupffer cells for two weeks, monitoring albumin production and basal IL-6 levels [26]. IL-6 administration has been used to mimic liver inflammation measured by the release of C-reactive protein and a decrease in cytochrome P450 enzymatic activity. Upon treatment with Tocilizumab, cytochrome P450 activity has been restored even in the presence of IL-6-induced inflammation, previously unachievable due to the challenges associated with maintaining traditional in vitro liver platforms in long term culture.

In a cross-species liver toxicity study, hepatocytes from rat, dog, or humans were co-cultured with liver sinusoidal endothelial cells in a tissue-tissue interface configuration [27]. This approach highlighted inter-species translational issues by testing a compound discontinued in Phase II human clinical trials due to human liver failure and premature death in one-third of the participants. A systematic toxicology review of the animal experimental data at the time concluded that the adverse reaction could not have been foreseen. Using rat and human liver-chips, the authors demonstrated a dose-dependent decline in human liver function, assessed by a decrease in albumin secretion, an increase in lipid accumulation, and an increase in liver injury markers, while rat liver function was unaffected.

### 3.2. Toward Body-on-a-Chip

OOAC systems over the past decade have been shown to have increased morphological and functional capacity within individual subsets of tissues, leading to a greater probability of capturing vital tissue responses to drugs. These platforms clearly have vast pre-clinical testing potential, with the potential of an individual OOAC likely to be compounded when multiple OOACs are coupled together to create integrated multi-organ or body-on-a-chip platforms. Yet, only 28% of OOAC developers in our survey indicated that they currently offer integrated OOAC models.

Many groups have developed multi-organ-on-a-chip systems, including liver-skin [28], intestine-kidney [29], and skin-hair models [30]. One of the first reported multi-OOAC platforms was developed by Michael Shuler et al. in 2004 [31], which linked multiple cell populations in independent compartments together in a single circuit with a “common blood surrogate”. This system provided the concept of obtaining entirely human responses to drug testing. More recently, their multi-OOAC system work has shown pharmacokinetic/pharmacodynamic (PK/PD) profiling of tamoxifen on MCF-7 breast cancer cells in the presence/absence of liver cells, while simultaneously monitoring the effect of tamoxifen on cardiac function [32].

OOAC models have the capacity to successfully recapitulate 3D cell environments by reducing tissue:cell culture media ratios, providing 3D scaffolds and matrices, and by facilitating different tissue-tissue interfaces. The Ingber research group reported a body-on-a-chip system utilizing tissue-tissue interfaces on opposite sides of a flexible porous membrane [19,33]. These devices typically contain a parenchymal/organ-specific tissue layer over which tissue-specific culture medium can be perfused, as well as a vascular endothelial tissue layer over which a common endothelial culture medium can be perfused from a central arteriovenous (AV) reservoir between vascular channels of independent chips, simulating systemic circulation. One study introduced an automated fluidic handling platform to maintain organ viability and function over 3 weeks in eight organ-chip models coupled via vascular endothelial-lined compartments [33]: gut, liver, heart, kidney, lung, heart, brain, blood-brain barrier, and skin. A follow-up study using the same system used models of four organs (gut, liver, bone marrow, and kidney) combined with in silico modeling to predict the PK of two drugs [19]: orally administered nicotine (via the gut-chip) and intravenously administered cisplatin (via the AV reservoir), which, when compared to human clinical and rodent in vivo data, provided clinically relevant data.

Another 3D body-on-a-chip cell culture model was introduced by Edington et al. and consisted of a custom well plate compatible with standard Transwell inserts and a pneumatically-driven pumping system, capable of sustaining 4, 7, or 10 microphysiological models (MPS). MPS functionality was assessed in each system after 2 weeks, 3 weeks, and 4 weeks, respectively, showing greater in vivo-like functionality over time. This study also attempted to scale the amount of flow to each compartment relative to the in vivo cardiac outputs to each tissue type using a flow partitioning strategy. Using the 7-organ format, diclofenac (DCF), a common non-steroidal anti-inflammatory drug (NSAID) with known experimental pharmacokinetics (PK), was administered to the apical compartment of the gut device. The PK of DCF and its metabolite 4-OH-DCF were determined by sampling medium from all OOAC compartments at 24 and 48 h time points and subsequent analysis by liquid chromatography/mass spectrometry (LC/MS). The platform demonstrated a greater in vitro PK predictive capacity when models were co-cultured than when models were studied in isolation.

Still, it is worth noting that the immense potential of body-on-a-chip platforms is in its infancy, and many logistical, mathematical, and biological challenges have yet to be overcome. Even with increasing numbers of individual organ models, perhaps body-on-a-chip platforms will only ever provide slightly more insightful snapshots of toxicity than existing technology and will always fall short in predicting the exact human response, owing to the immense complexity of the human body. Nevertheless, body-on-a-chip platforms may revolutionize pre-clinical testing.

OOAC developers and users were asked to rank specific design criteria for a body-on-a-chip system to be successfully validated, including in vivo vs. in vitro scaling of cell type composition, in vivo vs. in vitro scaling of cell surface area, cell volume, and cell to media volume ratio, metabolic scaling of interconnected OOAC devices, vascularization of interconnected OOAC devices, and common culture medium (similar to a blood substitute) for all interconnected OOAC devices. Respondents from the two groups scored with equivalent weights (Kruskal–Wallis, *p* = 1) the body-on-a-chip design criteria. Each of these criteria is discussed in Section 4.1 Scaling and Section 4.3.2 Vascularization and Universal Culture Media.

## 4. OOAC Challenges and Requirements Roadmap

While microfluidics and device manufactures have come a long way in enabling the development of OOAC technologies, progress and widespread adoption in drug discovery remain slow.

The survey results and conducted literature review resulted in a high-level chronological roadmap of OOAC challenges and requirements that, if considered carefully during development, may address many current shortcomings and lead toward broader adoption of OOAC devices by end-users (Figure 4). Individual phases of this roadmap are discussed in greater detail in the following sections.

### 4.1. Scaling

To maximize the predictability of OOAC models for pharmacological and toxicological research, test compounds should be applied to an interconnected collection of different organ models, allowing for xenobiotic metabolism between connected models [34,35,36,37,38]. Different delivery methods (e.g., oral, vaginal, pulmonary, anal, oral, or transdermal) of compounds will provide different challenges to pharmacological agents, such as digestion and transformation by local microbiota, host tissues, and host enzymes present in plasma [39,40].

The functional coupling of OOAC models imposes several important design considerations regarding metabolic profiling. The first consideration is related to the metabolic rate and biochemical activity of each OOAC model that is to be coupled with other OOAC models in order to maintain the physiological relevance of analytical results for normal and disease-specific models [41,42,43,44]. The consumption, metabolic conversion, and secretion of energy and respiration compounds, in addition to other relevant metabolites, are known to vary among different cell sources (i.e., immortalized cell lines, primary cells from donor tissue, human induced pluripotent stem cells), inter-laboratory handling and storage conditions, nutrient availability, and tissue maturity [45,46,47,48,49]. Further complicating matters, in vivo metabolic activity and secretion rates also vary based on the developmental stage (e.g., embryonic development, pregnancy), genetics, or disease status. In addition, Magliaro et al. outlined that correct scaling, especially in non-luminal models, such as brain and liver OOAC, is limited by oxygen diffusion in OOAC models, which should be considered during development [50]. In order to utilize OOAC models to predict the metabolic conversion of drugs or toxicants, models should be evaluated for their basal metabolic rate and scaled to other interconnected OOAC models accordingly.

A potentially problematic point for OOACs is a time-sensitive sampling of effluent as the perfusion speed of microscale devices is often very slow (typically 50–100 µL/h), unless the devices are accessible by micropipettes or are of sufficiently large volume to allow for faster collection of the required analytical sample volumes [19,33]. This is noteworthy because residual enzymatic activity in the samples should be stopped immediately after collection in order to minimize the post-collection degradation of unstable metabolites [51,52,53]. The speed of sample collection, in turn, has to be balanced with the effects of flow patterns and shear on the activation of biochemical pathways [54,55]. Previous studies using OOAC devices have shown the ability to analyze small volumes of “metabolite-rich” culture medium using untargeted systems-wide analyses (specifically, LC/MS-based methods) and provide longitudinal molecular mapping (hourly) from a liver-on-chip exposed to acetaminophen [56].

### 4.2. Device

In regard to device universality, 24% of OOAC developers reported that their company offers one universal device design to be used for different organs, another 24% reported that their company uses specialized device designs for specific organs, while the majority (44%) responded that their company produces both universal and specialized OOAC device designs.

#### 4.2.1. Material Selection and Standardization

Of the models currently under development, there is no standardized universal device design or device fabrication material. Prototyping of microfluidic devices for OOAC research can be achieved using any of the materials mentioned below; however, each has its advantages and disadvantages for cell culture and for data readouts. OOAC devices are commonly fabricated using polydimethylsiloxane (PDMS), a non-toxic flexible polymer with the capacity for cyclic stretching of cell culture surfaces, such as is used in some kidney [57] and pulmonary models [58]. The propensity of PDMS to absorb small hydrophobic molecules, such as tracers, drugs, and toxicants, is a drawback of the material [59]. Despite this concession, PDMS facilitates dynamic cell culture [60]. Other commonly used materials for microfluidic device manufacture, including glass [61], polyurethane [62], polymethyl methacrylate (PMMA), and other thermoplastics, are much less flexible but do not absorb hydrophobic molecules, perhaps leading to more reliable cytotoxicity drug testing data [63]. Research for sustainable alternative plastic for manufacturing (such as bioderived materials) and new manufacturing techniques obviously requires funding in order to limit and streamline disposable plastic waste management, improve accuracy and resolution of fabrication, and to support the creativity of the developers [64].

Standardization in this regard would simplify cross-lab validation of individual OOAC models, simplify the challenge of integrating OOACs with existing technology and equipment, and enable more rapid adoption of OOACs by the scientific community.

#### 4.2.2. Robustness and Reproducibility

Reproducible cell and extracellular matrix loading along with the prevention of air bubbles and ease of bubble removal inside OOAC devices were deemed the most important criteria for device robustness by both OOAC developers and end-users (Figure 5).

The underlying conditions that lead to air bubble formation inside microfluidics and various methods for prevention have been described by Pereiro et al. in detail [65]. Their review offers important design considerations for OOAC developers that include fluidic connections, chip structural design, and surface properties, as well as bubble trap options for an additional layer of protection against bubbles. Since this issue is rather common, preventative and corrective actions should be defined and communicated with end-users for a given OOAC platform.

Characterization of diffusion rates to evaluate kinetic homogeneity and reproducibility, especially when using membranes or 3D matrices, should be conducted early in the design and development stages.

### 4.3. Cellularization

#### 4.3.1. Cell Selection and Sourcing

For OOAC models to achieve broad adoption, translation of models to predictive clinical outcomes remains a challenge. The first consideration would be to ensure that all required functional cell types (e.g., for signaling, transport) are present at the correctly scaled ratios at the time of testing.

After identifying the necessary cell types, careful selections must be made in the sourcing of cells (see Section 4.1). Primary tissue and cell sources are oftentimes key to reproducing human-relevant healthy and disease models, yet this key component continues to require support in sourcing for both developers and end-users. Acquisition and isolation of primary cells from human tissue requires expertise, and isolated cells have a limited lifespan. Picollet-D’hahan et al. reviewed primary cells and different types of stem cells and outlined limitations in our understanding of signaling cascades to differentiate cells into functional organoids [66]. As one of the most important organs for drug metabolism studies, cell selection and sourcing for liver OOAC models have been given more extensive consideration. Ehrlich et al. assembled a set of what they refer to as minimum requirements for liver OOAC models, which include metabolic evaluation of the source cells with only HepaRG and Upcyte hepatocytes, satisfying metabolic requirements [20]. Induced human pluripotent stem cells (iPSC) have high potential by providing an unlimited source of cells and patient-derived material for precision medicine approaches. However, they often exhibit an immature phenotype, require validation of and consensus around differentiation methods due to line-to-line variability, and complex differentiation protocols continue to limit their widespread use as the preferred cell source [47,67,68,69]. Large-scale collaborative initiatives, such as HipSci (http://www.hipsci.org/) and STEMBANCC (https://stembancc.org/), help to provide key resources of cell banks with associated donor data and help drive the consensus required for the use of known donor cell lines for healthy or specific disease models.

#### 4.3.2. Vascularization and Universal Culture Media

Vascularization of OOAC models offers a way to increase physiological relevance via a specific barrier and transport functionalities, as well as interstitial flow modeling, specialized cell-cell interactions, metastatic models, inflammatory models, and drug- or chemical-mediated toxicity models. Various forms of vascularization have been implemented thus far—from separate endothelial cell-lined fluidic supply channels over bioprinted models to specialized angiogenesis chips [43,70,71,72,73]. Herland et al. [19] recently demonstrated quantitative pharmacokinetic responses to orally administered nicotine and intravenously injected cisplatin inside their interconnected, vascularized organ chips that had been perfused using a common blood surrogate [19]. The team prepared universal endothelial media by supplementing specific growth factors needed by the various interconnected OOAC models [33].

Additionally, Chen et al. highlighted the important balance between physiologically relevant culture media (blood surrogate) and volume-to-cell density scaling in order to avoid dilution of metabolites and potential off-target toxicities [74]. The tradeoff for this low media-to-cell ratio, however, seems to be a relatively short culture duration of only 72 h, making this model currently unsuitable for chronic exposure studies. In our survey, the developers and end-users had equivalent opinions on the needed culture or experiment duration. Thirty-six percent of developers indicated that their OOAC models are capable of continuous culture durations longer than 30 days, which meet the time indicated by users for chronic exposure testing.

### 4.4. Perfusion and Automation

Key issues raised by both developers and end-users included device robustness, reproducibility, affordability, and ease of use (Figure 6).

Compatibility with existing equipment and analytical tools are crucial for both academic and industrial users, where OOAC devices must be compatible with existing microscopes, incubators, and other downstream analysis techniques and tools. Such factors facilitate a greater need for automation in sampling/manipulation, such as media changes, drug additions, and perfusion to control fluid flow, as well as label-free real-time measurements of cell status and health, which address key issues in reproducibility and device validation.

Ehrlich et al. reviewed the critical relevance that perfusion metrics and shear flow stresses have on tissue nourishment, oxidative phosphorylation, metabolic function [20]. Similarly, shear flow stresses within certain ranges have been shown to regulate endothelial cell tight junction formation in blood-brain barrier models [75,76,77]. The perfusion rates should, therefore, be finely tuned together with the overall device design under consideration of nutrient consumption, waste production, shear flow stress, diffusion kinetics across membranes and throughout 3D hydrogels, metabolite degradation, and appropriate effluent production for analytical sampling.

For the prototyping of any individual OOAC model that is at some point to be integrated with other models, developers should also consider the necessary requirements for the device to be incorporated into a body-on-a-chip system. In our surveys, only 26% of developers currently offer specialized automated perfusion or fluid handling system for their platform, while only 8% of developers indicated that they currently offer a platform with specialized automation control software to schedule perfusion activities and/or sensor analyses (Figure 2F).

#### 4.4.1. Automation and Closed-Loop Control

The requirement for universal, sterile, and robust fluidic connections; automated operation of devices, including for media changes, pH testing, and drug/toxin addition; evaporation control; closed-loop sensor control of small culture volumes, allowing, for example, for automated pH adjustment based on sensor readouts were also deemed important by developers and end-users (Figure 5 and Figure 6). Interestingly, when asked to rank the most critical criteria toward the broad adoption of OOAC technology, real-time monitoring was considered more important for users in pharmaceutical companies than from academia.

Automated closed-loop monitoring and control of respiratory (oxygen), energy metabolism (glucose/lactate), acidification (pH), and other key functional metabolites would enable a large-scale culture of OOAC devices in pharmaceutical and toxicological testing environments while supporting regulatory and quality standards in regard to monitoring and recordkeeping. Several platforms for simple in situ monitoring of OOAC already exist [78,79,80]. Real-time, closed-loop control could trigger changes in environmental oxygen concentration or the addition of fresh culture media to replenish exhausted glucose/lactate levels. This type of automation would be especially useful for microscale volumes that are more easily perturbed by external influences, oftentimes difficult to monitor via stain-free microscopy, and have little analytical sample volume to spare. As such, Kieninger et al. reviewed electrochemical and optical microsensors to monitor the metabolic activity of microfluidic cultures, while Santbergen et al. assembled an overview of real-time, automated platforms, specifically for OOAC models [80,81]. Young et al. provided a perspective of integrated microfluidic sensors that would move toward autonomous decision-making, for example, to evaluate OOAC culture maturity for testing and quality control purposes [82]. Yet, while a variety of microfluidic sensor platforms are already in existence, more advanced closed-loop sensor control platforms would greatly reduce the monitoring and handling burden for laboratory scientists (end-users), aid quality control, and increase the potential for scale-up.

#### 4.4.2. Online vs. Offline Integration

Online multi-organ system interactions encapsulate any system whereby individual organ modules possess a direct fluidic connection with one or more individual OOAC models in a self-contained perfusion system. Online connections are applied in several of the aforementioned platforms and have acted as PK predictors [19,38]. Culture media with or without drug or toxicant additions can be absorbed and transformed by exposed cells in real-time at prescribed perfusion rates and consistent pressure before being passed to a downstream organ model.

Contrary to online integration, offline connections of multi-organ modules are an indirect approach, whereby media effluent from one organ system can be collected, transferred, and later perfused through a second, independent organ system. Offline connections allow the user an opportunity to analyze medium composition after it has been conditioned and functionally transformed by the first organ model before this medium is transferred to the next model. It also facilitates inter-institutional collaboration, as shown by the work of Vernetti et al., who brokered collaboration between different universities [83]. In this work, a multi-organ microphysiological system was devised to mimic the physiological passage of metabolites through the body via the route of the intestine, liver, kidney proximal tubule, and blood-brain barrier. Terfenadine, a cardiotoxin with inhibitory effects on skeletal muscle, showed a markedly reduced effect on skeletal muscle using medium pre-incubated in the liver model, suggesting liver metabolism being crucial to avoiding adverse cytotoxicity.

A central issue that may arise with offline connections is the feasibility of industrialized scaling of body-on-a-chip platforms due to the manual labor involved in setting up multi-organ platforms with offline connections. As such, offline models could pose obstacles to the broader adoption of OOAC models. Especially for chronic exposure, testing the effort required for manual manipulations can be substantial and generally tends to be more error-prone unless automated fluid handling robots are utilized, which, in turn, require the design inclusion of ports for robotic fluid handling access. Novak et al. developed a programmable hybrid platform that integrates robotic fluid handling with computer-controlled peristaltic perfusion to fluidically couple multiple independently perfused, vascularized organ models [33]. The team used scored plate sealing films to minimize evaporation from robot-accessible fluid ports and was able to demonstrate metabolic linking of organ models over 3 weeks. Interestingly, the system can be configured for drug dosing or toxicant exposures via oral (intestine), intravenous (vascular), dermal (skin), and airway (lung) routes.

### 4.5. Validation

Approximately one-third of survey respondents (36% of end-users and 37% of developers) agreed that OOAC models need better validation methods to be more broadly adopted. An additional 18% of end-users and 20% of developers indicated that formal assessment and standardization would increase credibility, while 17% of developers highlighted the need for additional cell validation research. Robustness and reproducibility are issues raised by 13% of OOAC developers. Both OOAC developers and end-users in our survey agreed that molecular transport, metabolism, and inflammatory responses are the most important biological functions that OOACs should be capable of replicating and, as such, are important toward the broad adoption of OOAC (Figure 7).

#### 4.5.1. Metabolic Functionality and Response Assessment

One way to assess baseline metabolic functionality and response to challenge agents, such as xenobiotics or toxicants, is a multi-omics analysis of cells or OOAC device effluent. The multi-omics analysis integrates data derived from different “-omic” disciplines (e.g., genomics, transcriptomics, proteomics, metabolomics) and enables dynamic and powerful characterization of human health and disease [84,85]. Gutierrez et al. developed automated sample preparation protocols for high-throughput multi-omic analyses, including transcriptomics, proteomics, and metabolomics, for efficient acquisition of large, multi-condition, multi-timepoint datasets. Uhlén et al. integrated proteomics and quantitative transcriptomics with immunohistochemistry to assemble a tissue- and organ-level overview of the human proteome and compare expression patterns between normal tissues and corresponding cancer cell lines [84,86,87].

Metabolites present the end-products of the biochemical reactions within an organism, linking gene expression, protein function, and environmental influences. Broadscale molecular analyses have the potential to measure all secreted and consumed small molecules and identify novel compounds or pathway perturbations, including small molecules (metabolomics), fatty acids and lipids (lipidomics), carbohydrates (glycomics), and amino acids. Metabolites and metabolic pathway perturbations have been shown to be promising indicators of various diseases [88]. Metabolomics also enables the analysis of inflammation-related compounds, the co-evolution of host-microbiome interactions, and endogenous compounds, such as environmental exposures (exposome) or diet-derived chemicals. Targeted assays can subsequently be applied for validation and absolute quantification of specific compounds or biomarkers of interest. Such small molecule and multi-omic profiling can be accomplished using small sample volumes (<60 µL) [37,84,89,90,91]. This allows for non-endpoint characterization of basal, diseased, drug-treated, or toxicant-challenged conditions over time as has been successfully conducted by multiple research groups using integrated multi-organ-on-a-chip models [83,92,93].

Another important consideration should be given to enzymatic activity within the OOAC model and circulating media. In addition to tissue-specific enzymes that may transform drugs and toxicants, the microbiome may transform xenobiotics prior to absorption, and extracellular vesicles secreted into the bloodstream in vivo can also affect pathophysiological processes [94,95,96,97,98]. These biotransformation processes have the ability to activate or deactivate xenobiotics, making them reactive or non-reactive. Cytochrome P450 enzymes (CYPs) are the most common enzymes involved in various xenobiotic Phase I reactions, such as oxidations, reductions, and hydrolysis reactions [99]. They can be found primarily in the liver but also in various other organs, such as the gut, brain, kidneys, and lungs [100,101]. Notably, CYPs also circulate in the bloodstream in CYP-containing extracellular vesicles [102].

To harness the full potential and value of OOAC models, they should, therefore, be designed with future interconnectivity and inclusion of the microbiome metabolome in mind [39,95,96].

#### 4.5.2. Microbiome-Host Co-Metabolism

Microbiota colonizes various areas of the human body, including dermal, oral, pulmonary, intestinal, urethral, vaginal, and mammary gland tissues, and are influenced by external perturbations, such as pharmacological agent or environmental toxicant exposures [103,104,105,106]. Rapid advancements of microbiome research in recent years have led to a better appreciation of the host-microbiome co-metabolism and its feedback loops in health and disease. As such, important areas of crosstalk between the human microbiome and its host enabled by metabolites have been identified for various organs, including the gut, brain, heart, lungs, liver, and kidney [107,108,109,110,111]. However, various disease-associated interactions and pathways have already been associated with forms of dysbioses and identified for inflammatory bowel diseases, cancer, neurodegenerative diseases, and preterm birth [112,113,114,115,116,117,118]. In addition, the gut microbiome has been found to alter xenobiotic metabolism, drug efficacy, and toxicity [39,119,120,121]. One of the modes in which the microbiome plays an active role in drug metabolism, drug efficacy, and toxicity is via its effects on CYPs [39,120,122]. These interactions demonstrate the importance of including gut-on-a-chip models with microbiomes in epidemiology, pharmaceutical, and toxicological research using organs-on-a-chip. The European Research Council is funding such a microbiota-multiorgan-on-a-chip project with the aim of providing new insights into neurodegeneration [123]. Meanwhile, several groups have successfully published gut-on-a-chip models that incorporate microbiome cultures [124,125,126]. As the field of microbiome research continues to acquire new analytical technologies and knowledge, it will be important to integrate those findings into OOAC research.

#### 4.5.3. Disease Model Validation

The importance of disease model characterization and validation was reflected equally across developers and end-users. Understanding the limitations or suitability of any ‘on-chip’ disease model is highlighted as a key requirement, resulting from both biological and technical restrictions. The absence of underlying biological functions like immune, metabolic, or inflammatory responses complicates translatability to clinical disease. It is, therefore, unsurprising that the respondents in the survey were united in their validation needs for single defined organ/disease models and their appreciation that only a subset of functions may be reproduced (Figure 7B). Furthermore, a recapitulation of prolonged disease processes (e.g., fibrosis) or opportunities for chronic dosing into timescales consistent with OOAC technology is a significant desire. Beyond 30 days is a demand from both testers and developers for the development of slow diseases or mixed pathologies, which brings challenges around ensuring prolonged culture viability and sterility, while stimulating the requirement for label-free, real-time, continuous readout technologies. As the combination of multi-OOAC devices is progressing, the accumulation of the deficiencies associated with each model per se brings further biological and translational complexity.

Consequently, developers and end-users need to carefully consider what organ functionality and diseases can be recapitulated in each OOAC model and what validations have already been proven for the specific model. Encouraging examples of good translation for OOAC are building, in an on-chip model of Bart’s Syndrome, diseased cardiomyocytes, showing structural and functional deficiencies reflective of the clinical situation [127]. The combination of multiple cell types and vacuum-assisted mechanical force in a lung-on-a-chip model has produced organ-level responses, including neutrophil responses and IL-2-mediated edema [128]. To understand and validate OOAC utility for specific disease research, a comprehensive and unbiased assessment of functionality is recommended, ideally using clinically relevant biomarkers from clinically relevant cells. Newer omics-based technologies mentioned previously are well placed to achieve this where gene expression or proteomic response of up to 800 genes can be assessed in samples at a scale consistent with OOAC systems. The effects observed at this level of detail may both inform the biological relevance of the model and address limitations due to technical aspects. Towards body-on-a-chip models, those issues associated with relative inter-organ scaling, interdependent organ functionality, the identification of a suitable common media, and an understanding of required relative flow rates need to be considered.

#### 4.5.4. Pharmacological and Toxicological Research Requirements

The opportunities associated with OOAC technology are attractive for pharmacological and toxicological compound assessment. In projects aimed at finding an efficacious new drug, the technology enables the in vitro assessment of single and connected interdependent organ biology, such as glucose and insulin relationships between liver and beta cells of the pancreas. While acknowledging that these systems do not recapitulate all biological functions, their added value is beyond existing models and an alternative to pre-clinical in vivo testing. Importantly, the cell models should reflect the chosen patient population that is the target for the therapy. The opportunity to better model drug effects in OOAC exists if the liver or other highly metabolically active components are part of the interacting system (as indicated earlier). Multi-OOAC platforms are, therefore, highly relevant for ADME-toxicity evaluations [18].

Furthermore, scaling of drug doses, effective single or repeated dose effects, and relevant exposure duration with associated compound detection approaches will permit modeling of plasma concentrations, half-life, tissue distribution, etc., based on extensive experience in measuring these parameters from in vivo drug metabolism and pharmacokinetics (DMPK) studies.

The survey’s developer respondents identified pharmaceutical and contract research organizations (CROs) as target customers. An appreciation of the positioning of OOAC approaches in these environments is necessary to inform marketing approaches. The barriers to adoption are not only the relevance of biological models but include throughout compatibility with established workflows and costs. For the pharmacological validation of OOAC disease models, the recapitulation of clinically observed compound effects is highly prized. However, for a novel drug target, this clinical data is, by definition, missing. Conversely, for toxicity assessment, many drugs are well annotated for a specific target organ toxicity (e.g., hepatotoxins). Identification of these toxins in an OOAC approach is, therefore, dependent on the availability of the corresponding functionality within the model. This may be present in primary cells, which have limited supply, or absent in iPSC-derived cells, which can be massively scaled. In addition, iPSC-based models enable sophisticated engineered mutations and isogenic controls, which may increase the confidence of observed compound effects compared to the more complex primary patient material. However, where target toxicity is unknown, there remains an almost insurmountable task to represent all possible permutations, such as different targets, metabolic pathway perturbations, etc., which can occur in vivo. Taken together, these conclusions indicate that efficacy projects will benefit ahead of toxicological research in the short term.

## 5. Translation

### 5.1. Technology Readiness Level

Technology readiness levels (TRL) shown in Table 1 are used to indicate technological maturity and were initially introduced by the U.S. National Aeronautics and Space Administration in 1974. They were later adopted by the U.S. Department of Defense, European Space Agency, ISO 16290:2013 standard, and the European Union’s Horizon 2020 program [129].

When asked to estimate the maturity of the OOAC field by assigning a TRL, the majority of respondent end-users (39.3%) assigned “TRL 4—Technology validated in the laboratory”, while equal percentages of OOAC developers assigned TRL 4 and “TRL 7—System prototype demonstration in an operational environment” (24%) (Figure 8A). The higher TRL assignments made by OOAC developers than end-users may be due to proprietary information about existing industry collaborations and overall familiarity with the technology as the largest percentage of participating end-users has only worked with OOAC for two years or less. However, from an end-user’s perspective, it may certainly be frustrating that even OOAC developers do not rate their commercially available products at a TRL of 8 or 9. This indicates that the technology offered is still in development and not in its final form: only in the last couple of years, the emerging OOAC companies have established partnerships and collaborative relationships with pharmaceutical companies to accelerate the development of OOAC technology and testing [27]. Peer-reviewed publications of analytical results from these operational environments have reported the first successful applications in operational conditions (TRL 7) and confirmed the potential for OOAC to improve the process for predictions of adverse drug reactions before drug candidates enter clinical trials [131,132].

### 5.2. Existing Standards

As a new, multidisciplinary concept, an OOAC system involves materials, biological samples, and integration with other technologies. Based on its definition, it has the potential to become a key component of the drug testing and drug screening process, which is a highly regulated field. The survey thus aimed to understand the users’ level of awareness and the approach of established companies with respect to potentially applicable classification and regulations. To our knowledge, OOAC does not belong to any specific device classification by the main regulatory bodies, such as the U.S. Food and Drug Administration (U.S. FDA), the European Medicines Agency, and/or the Medicines and Healthcare Products Regulatory Agency in the United Kingdom. However, the annual reports of the FDA’s Office of Clinical Pharmacology, the Center for Drug Evaluation and Research (CDER), and the Division of Applied Regulatory Science started to report in 2017 their first evaluation and co-development of OOAC (microengineered human cellular systems, microphysiological systems) as new revolutionary models to improve the assessment of drug safety and the general understanding of the diseases caused by bacteria and chemicals [133,134,135].

Many existing quality management and regulatory systems may have applications to OOAC manufacturing and validation and could be applied by the OOAC community until specific standards are introduced. The international organization of standardization has devised and regularly updates an internationally agreed-upon quality management system (ISO 9001:2015). This system can be applied to any type of organization and demonstrates the ability to provide products and services of consistent quality by defining requirements and guidelines for documenting information, process planning, leadership and resource management, performance evaluations, and preventive/corrective actions. ISO standard 13,485 is a quality management system similar to ISO 9001 but outlines requirements specific to the medical device industry, including design and development planning, inputs, outputs, review, verification, validation, transfer and control; cleanliness, contamination control, and sterility; as well as monitoring, audits, and traceability, among others [136].

Standards published by the U.S. Centers for Medicare and Medicaid Service, called clinical laboratory improvement amendments (CLIA), for clinical diagnostics or personalized medicine may be applicable to OOAC models. CLIA regulations apply to facilities aiming to provide diagnostic, preventative, disease treatments, or health assessments of humans. While the exact certification requirements depend on the type of lab test performed and its complexity, they generally include implementation and auditing of a quality management system to ensure specimen and data integrity, proficiency testing, personnel competency assessments, quality assessments, as well as procedures for maintenance, quality checks, calibrations, and corrective actions.

The U.S. FDA has developed regulatory guidelines for Good Laboratory Practice for Nonclinical Laboratory Studies (21 CFR Part 58), which includes preclinical toxicology studies [137]. These regulations require the implementation of a quality assurance unit, study director, and specific testing facility management and outline special requirements for specimen and data storage; study protocols and standard operating procedures (incl. for corrective actions); tracking of reagents and solutions (incl. batch numbers and expiration dates); written records for equipment maintenance, cleaning, calibration, and design; handling and archival of samples from each processing batch for studies lasting longer than 4 weeks; as well as reporting of results, data storage and retrieval, and records retention.

The U.S. FDA also developed a Qualification of Drug Development Tools Qualification Program [138] (FD&C Act Section 507), which includes the Animal Model Qualification Program (AMQP), Biomarker Qualification Program (BQP), which are potentially applicable to OOACT [139]. The AMQP qualification evaluates the fitness of the model for a specific use, with characterization of the challenge agent(s) and exposure, primary and secondary endpoints, triggers for intervention, and key disease values to be replicated for quality assurance and control.

Notably, our survey demonstrated that 44% of OOAC developers and 50% of end-users were unfamiliar with any of the existing regulatory standards and guidelines outlined above (Figure 8B). According to our survey, the standards most used by OOAC developers are ISO 9001:2015 (22%), FDA 21 CFR Part 58 (22%), and FDA FD&C Act Section 507 (16%). Only 2% of OOAC developers surveyed noted the use of ISO standard 13,485 for medical devices, while the application of CLIA standards (4%) and FDA AMQP (4%) was similarly low.

While none of the above outlined regulatory standards and quality management processes may be a perfect fit for OOAC technology, familiarity with these frameworks would certainly ensure reliability, reproducibility, and completeness of the experiments and data generated using OOAC models. By referring to common protocols for testing, collection, and assessment of data, it would be possible to generate new and more accurate standards and policies. This new level of quality and regulation assurance would positively impact the adoption of the technology in the scientific and industrial communities and would aid the classification of new OOAC products in the market. As for any other emerging technology, compliance with quality standards and certifications is not always contemplated in academic research, at the proof of concept stage or low TRLs, because of costs associated with evaluation and improvement of existing processes, qualification of equipment, reference materials and procedures, to name a few. However, for emerging small companies, compliance with quality and regulatory standards for validation of methods, management of risks and requirements, documentation and traceability, personnel training, and inter-laboratory comparisons would support robust and reproducible development and fabrication of OOAC devices and platforms and inevitably boost the adoption and establishment of OOAC models as the new standard for highly-regulated end-users, such as the pharmacological and clinical industries.

In the absence of specialized regulatory guidelines, existing standards could be adopted for hardware components and processes used in the OOAC field, such as the use of analytical-grade tubing and high-pressure fluidic fittings from ISO-certified suppliers. Learning to develop corrective and preventative actions (CAPA), as outlined by the U.S. FDA and Medical Device Innovation Consortium in a recent white paper, could be similarly beneficial [140].

### 5.3. End-User Support

Developers and end-users were also in agreement about the most important end-user support categories toward broad adoption (Figure 9).

Standard device operating protocols and rigorous end-user training were most important, followed by advanced device operating protocols for specialized applications, such as disease models, support in cell sourcing and cell selection, in-house disease model validation, and consultations for the design of experiments. One of the key end-user support areas selected by respondents is contract research services (CRS). These may be seen as a gateway into the pharmaceutical industry as they offer the ability to trial new innovations and technologies at a more widespread scale without steep learning curves required for the operation of complex OOAC models and necessary quality management practices [141]. Overall, challenges, such as device standardization, detailed use cases for OOAC systems to replace or support existing drug discovery assays, and costs of implementation and specialist skills required for model set up may impede the rapid uptake of such technologies. However, continued collaborative partnerships between industry, academia, and CRS organizations are essential in the development and evaluation of OOAC technology. As discussed in the previous sections, several aspects require technological development and, thus, financial support. In terms of broader technology areas that require additional research, users from pharmaceutical companies ranked cell sourcing, validation, and specialized automation higher compared to academic users. Interestingly, device sustainability was considered a big issue in academia more than in industry.

## 6. Discussion and Conclusions

Our survey reached a small fraction of the global OOAC field. However, it confirms that OOAC research and development are mainly characterized by experienced scientists, young users, new businesses, and mostly dominated by small and medium-sized enterprises and academic researchers.

Overall, the survey data showed a high consensus regarding the challenges and technological bottlenecks faced by the community, requirements for new technologies and services, and the need for more detailed validation of individual and interconnected models in order to reach more advanced technology readiness levels required for broader adoption.

Based on the survey, many developers and end-users agreed on a low technological readiness level, suggesting that the field is still far from thorough and consistent validation, and thus there is a great need for further development and testing activities to reach broad adoption and acceptance of OOAC by large pharmaceutical companies and regulatory agencies.

Broad multidisciplinary expertise is required to advance the field to higher technology readiness levels as consensus among respondents shows that several design requirements still need to be addressed to guarantee OOAC robustness and reproducibility in order to meet the standards in commonly highly regulated environments to predict drug responses and toxicological outcomes (Figure 5, Figure 6, and Figure 7B).

The survey confirmed the need for additional focused validation efforts and the commercial interest in numerous technology areas, such as integration of sensors and hardware for enhanced control of the microenvironment and for higher automation of procedures (e.g., for cell or liquid loading, drug injection, and sampling). A key challenge is represented by the innate variety of cell sources and the complexity of the OOAC models. This complexity is reflected by the need for training and guidance to end-users. Operational reliability, accuracy, and precision need to be assessed to validate the OOAC value proposition. The important drivers for global organ-on-a-chip market growth have been the early recognition of its potential in drug toxicity testing and the technological advancements that followed. As such, the OOAC market has gathered many players from different horizons, and it is evolving continuously due to rapid technological advances and the field’s strong multidisciplinary attractiveness. Based on the variety of models, mature or in development, a universal device design seems unlikely. Nevertheless, the authors expect that the most widely adopted, flexible, thoroughly characterized, and validated platforms will ultimately be used to set new regulatory standards and force other developers to revise their platforms around this gold standard.

In this regard, the OOAC community is still maturing, and thus there is much room for growth, including official regulatory guidelines, specifically for the validation of OOAC technology for a variety of potential applications, including pharmacological and toxicological testing. It is good that standards are not introduced too early in the development process so as not to hinder technological advancement. However, while this may be beneficial to the open development and market introduction of a broad variety of OOAC constructs and supportive technologies, the lack of regulatory standards risks to limit not only the application of OOAC to support pharmaceutical and chemical safety assessments but also the trust end-users are able to place in the technology, which could, in turn, foster broader adoption.

While academic research groups may choose not to be fully compliant with regulatory and quality standards, their research and development would benefit from greater familiarity with industry standards for quality management, such as ISO:9001 and ISO:13485, just as much as any new and established OOAC enterprises.

Finally, the results of our survey show greatly overlapping interests and a willingness to co-develop existing OOAC platforms, ancillary technologies, and tools required for the use of OOAC devices. This active and collaborative research and commercial effort could potentially speed up the advancement of OOAC in the market, enable broader adoption and acceptance of OOAC, and continue to support the growth of the market in the years to come.

## Figures and Tables

**Figure 1 bioengineering-07-00112-f001:**
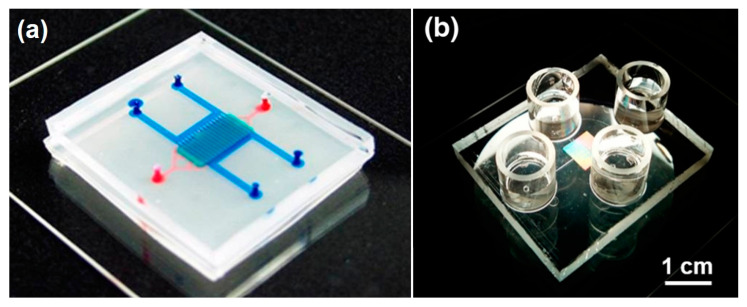
Examples of organs-on-a-chip models, recreating (**a**) the blood-brain barrier and (**b**) the perivascular environment in the human endometrium (Adapted and Reprinted from [1], with the permission of AIP Publishing and from [2], both licensed under a Creative Commons Attribution—CC BY license).

**Figure 2 bioengineering-07-00112-f002:**
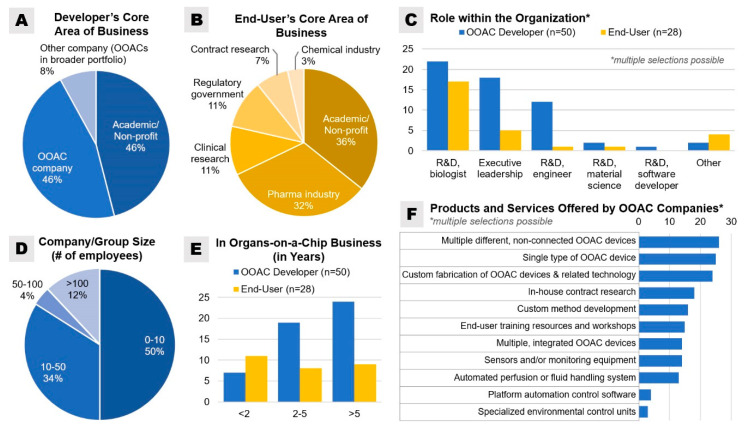
Overview of organ-on-a-chip (OOAC) developer and end-user survey respondents: (**A**,**B**) areas of business (single selection), (**C**) role(s) within their organization (multiple selections possible), (**D**) OOAC developer’s company or research group size (single selection), (**E**) respondent’s number of years in the OOAC business (single selection), and (**F**) an overview of the products and services offered by responding OOAC companies.

**Figure 3 bioengineering-07-00112-f003:**
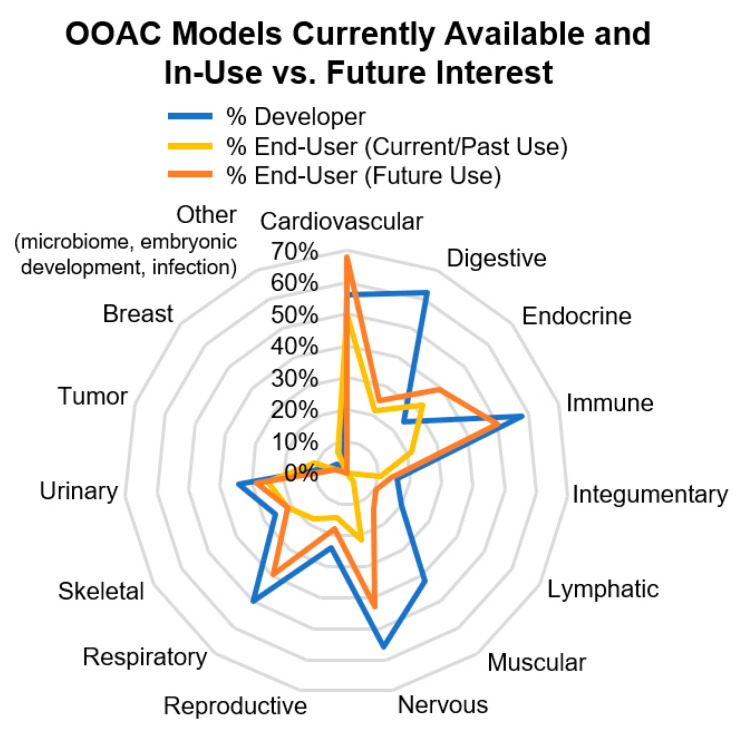
Overview of available and desired OOAC models indicated by survey respondents. The responses given by OOAC developers and end-users were statistically different (*t*-test, *p* < 0.001 comparing developers vs. current and past users, *p* < 0.005 comparing developers with future interests), while the end-users’ past, current, and future interests were not significantly different.

**Figure 4 bioengineering-07-00112-f004:**
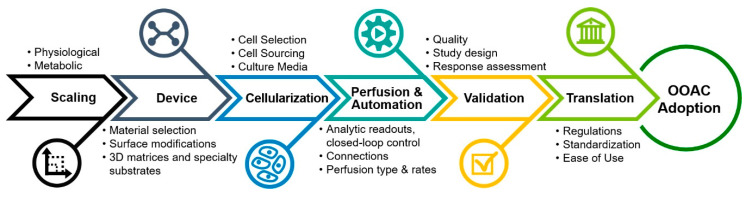
High-level roadmap of OOAC challenges and requirements, which begins with scaling considerations and progresses through development phases, encompassing device development, cellularization, perfusion and automation, and validation requirements before reaching a translational stage that enables specific regulations, standardization, and provides a level of ease-of-see that can lead to greater OOAC adoption.

**Figure 5 bioengineering-07-00112-f005:**
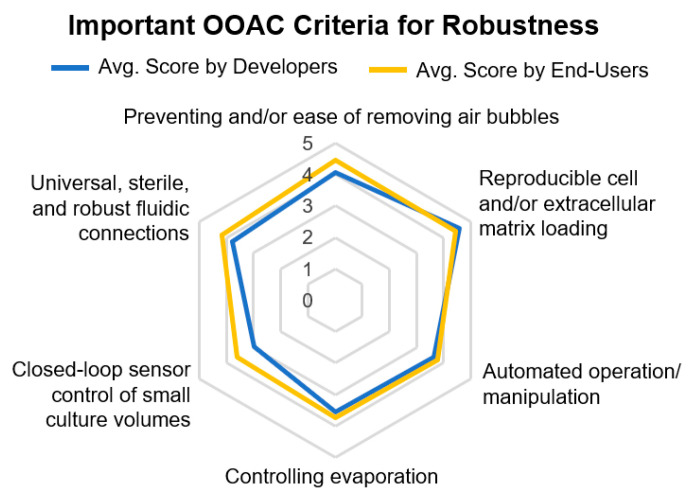
A radar map, showing a consensus among survey responses regarding criteria important for OOAC robustness. Examples given to respondents were Luer-Lok for fluidic connections, media changes, pH testing, the addition of drug/toxin for automated operation/manipulation, and automated pH adjustments based on sensor readouts for closed-loop control. Responses were collected on a consistent Likert rating scale from “1—not important at all”, “2—somewhat important”, “3—neutral”, “4—somewhat important”, to “5—very important”. OOAC developers and end-users ranked the criteria for robustness (*p* > 0.05) of the new models in equivalent order (*p* = 1, Kruskal–Wallis).

**Figure 6 bioengineering-07-00112-f006:**
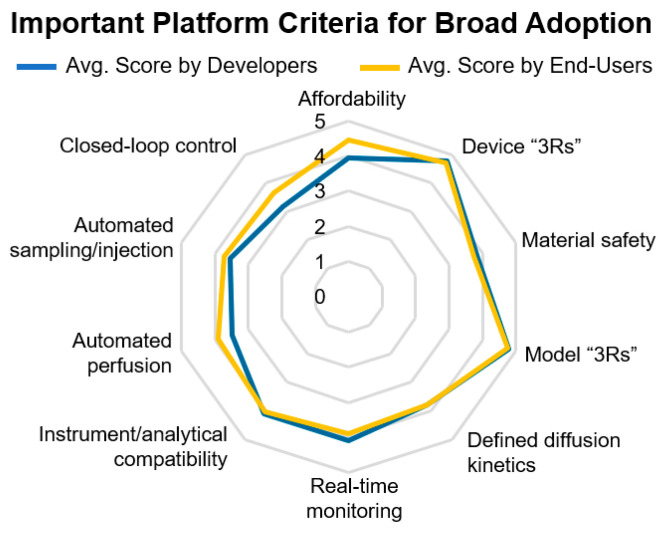
Overview of critical platform criteria toward the broad adoption of OOAC technology. Abbreviated in this figure as Device “3Rs” and Model 3Rs” are robustness, reproducibility, and reliability of each. Examples for automated perfusion and automated sampling/injection included pumps, valves, and robotic pipettes, while examples of closed-loop control included automated pH and oxygen measurements and subsequent adjustments. OOAC developers and end-users ranked the criteria for the broad adoption of the new models in equivalent order (Kruskal–Wallis, *p* = 1).

**Figure 7 bioengineering-07-00112-f007:**
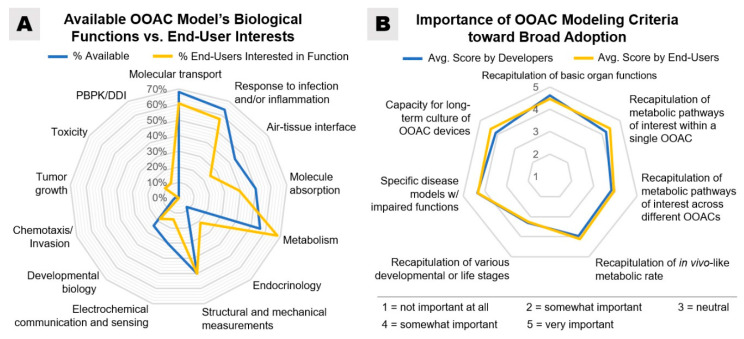
(**A**) A radar map, showing survey responses comparing biological functions available for developers vs. biological functions desired by end-users. Respondents were able to select multiple-selection answers. OOAC developers and end-users ranked similarly to the essential biological functions to be modeled in an OOAC (Kruskal–Wallis, *p* > 0.1). (**B**) A radar map, showing a consensus among OOAC developers and end-users regarding the importance of certain modeling criteria toward the broad adoption of OOAC technology. Responses for this question were collected on a consistent Likert rating scale from “1—not important at all” to “5—very important”, as shown. Cardiac cell contractions were given as an example for recapitulation of basic organ functions, while the capacity for long-term culture could benefit the evaluation of chronic exposures and tolerance build-up.

**Figure 8 bioengineering-07-00112-f008:**
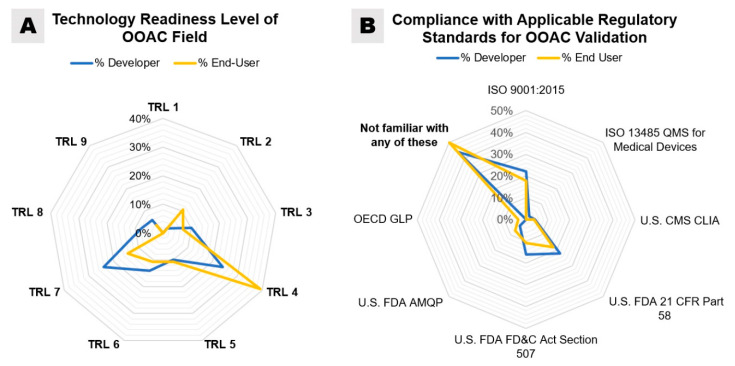
(**A**) A radar map, comparing survey responses by OOAC developers and end-users and their estimation of technology readiness levels (TRL) of the OOAC field. (**B**) A radar map, showing overlap among users’ and developers’ responses regarding compliance with existing quality and regulatory standards. Multiple selections were possible. Answers from the two groups of respondents were not significantly different (*p* > 0.05).

**Figure 9 bioengineering-07-00112-f009:**
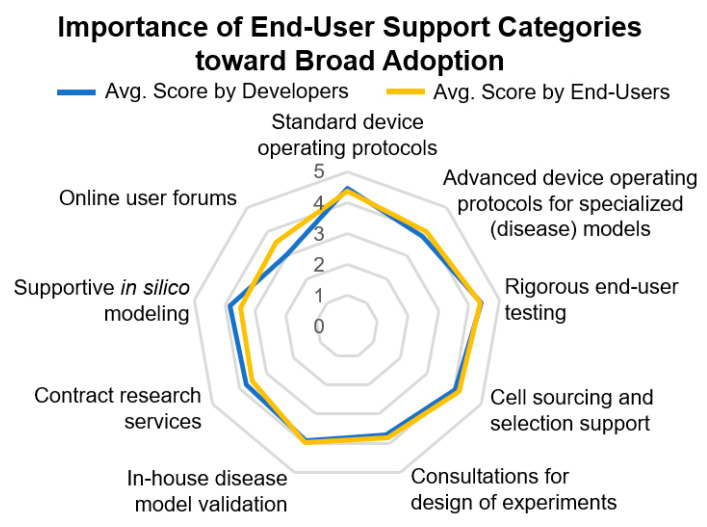
A radar map, showing a consensus among survey responses regarding the importance of end-user support categories toward broad OOAC device adoption. Responses were collected on a consistent Likert rating scale from “1—not important at all”, “2—somewhat important”, “3—neutral”, “4—somewhat important”, to “5—very important”. OOAC developers and end-users ranked the criteria for support services or trainings (*p* > 0.05) of the new models in equivalent order (*p* = 1, Kruskal–Wallis).

**Table 1 bioengineering-07-00112-t001:** EU Horizon 2020 technology readiness levels (TRL) [130].

Level	Description
TRL 1	Basic principles observed, the start of scientific research
TRL 2	Technology concept formulated, no to very little experimental proof of concept
TRL 3	Experimental proof of concept conducted
TRL 4	Technology validated in the laboratory
TRL 5	Technology validated in a relevant environment (industrially relevant environment in the case of key enabling technologies)
TRL 6	Technology demonstrated as a fully functional prototype in a relevant environment (industrially relevant environment in the case of key enabling technologies)
TRL 7	System prototype demonstration in operational environment
TRL 8	System complete and qualified
TRL 9	Actual system proven in an operational environment (competitive manufacturing in the case of key enabling technologies or in space)

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
