# Peer review of "Translational Roadmap for the Organs-on-a-Chip Industry toward Broad Adoption"

_bioengineering, 2020, doi:10.3390/bioengineering7030112_

Round 1

Reviewer 1 Report

Not crucial, but a few figures and/or photos depicting OOACs would be nice for the non-specialists.  

Author Response

“Not crucial, but a few figures and/or photos depicting OOACs would be nice for the non-specialists.”  

We thank the Reviewer for the positive comments on the originality and well-structured manuscript.

We included Fig. 1 as example of previously published organ-on-a-chip to address this request.

Reviewer 2 Report

Interesting topic of study. It will be great if the data can be presented in a more exciting manner. 

  • Many of the radar maps show a consensus among the developers and the users. I wonder how they will look if the data are plotted as academics vs industry respondents. Particularly when the authors have almost equal number of respondents from both. This will be essential to see if there are some areas which the academics work on but are not considered very important by the pharma companies. Can it have any impact to identify industry funding opportunities?
  • In general, can the authors classify the respondents better, to bring out more information in the data presented? Unfortunately, showing the average score only is not very exciting.
  • Expand ADME, IL-6, TNF-alpha at their first usage.
  • Line 347: ‘criterium’ is incorrect.
  • Please rewrite the sentence from line 177 through 179.
  • Line 249: Delete “Scaling” and “Vascularization…”. Not required if the section numbers are mentioned.
  • Section 3.1 first paragraph: Please expand on the liver OOAC models with appropriate references.
  • Section 4.1 first paragraph: Please add appropriate references.
  • Section 4.3.1: Please add appropriate references.
  • Section 4.4, second paragraph: Please add appropriate references.
  • Page 14: Please rewrite: “Consequently, developers and end-users need to be realistic and specific in any claims of disease recapitulation and not seduced by ‘hype’.”
  • Page 14: “Towards body on chip models…. big data solutions”. Please elaborate or add appropriate references.

Author Response

“Interesting topic of study. It will be great if the data can be presented in a more exciting manner.”

We appreciate the comment of the Reviewer and we thus revised the manuscript, addressing the point raised below and by the other two Reviewers. We hope these changes make the manuscript more readable and appealing.

  • “Many of the radar maps show a consensus among the developers and the users. I wonder how they will look if the data are plotted as academics vs industry respondents. Particularly when the authors have almost equal number of respondents from both. This will be essential to see if there are some areas which the academics work on but are not considered very important by the pharma companies. Can it have any impact to identify industry funding opportunities?”

We thank the Reviewer for suggesting this further analysis. To develop this point we selected two specific questions in the Survey that highlighted different opinions of users from pharma vs academics. Specifically in Question 13 (Please rank the following OOAC platform criteria for criticality toward broad adoption of OOAC technology.), real-time monitoring is considered more important for users in pharmaceutical companies than from academia. In Question 21 (Please rank which of the following broader technology areas require additional research) users from pharmaceutical companies ranked cell sourcing, validation and specialized automation higher compared to academic users. Interestingly device sustainability is considered as a big issue more in academia than in industry. We included in section 4.4.1 and in section 5.3 two sentences to comment these findings in the main text of the manuscript and we added in the Excel file the pulled data corresponding to these.

  • “In general, can the authors classify the respondents better, to bring out more information in the data presented? Unfortunately, showing the average score only is not very exciting.”

We have now included a new Excel file, with a more readable version of the survey results, highlighting the answers received from the respondents of the different categories. From these data we derived the score, as described in the methods and as calculated in the Excel file.

We finally thank the Reviewer for highlighting specific words and sentences to be modified. We addressed the points raised below and tracked the changes in the revised version of the manuscript:

  • Expand ADME, IL-6, TNF-alpha at their first usage.
  • Line 347: ‘criterium’ is incorrect.
  • Please rewrite the sentence from line 177 through 179.
  • Line 249: Delete “Scaling” and “Vascularization…”. Not required if the section numbers are mentioned.
  • Section 3.1 first paragraph: Please expand on the liver OOAC models with appropriate references.
  • Section 4.1 first paragraph: Please add appropriate references.
  • Section 4.3.1: Please add appropriate references.
  • Section 4.4, second paragraph: Please add appropriate references.
  • Page 14: Please rewrite: “Consequently, developers and end-users need to be realistic and specific in any claims of disease recapitulation and not seduced by ‘hype’.”
  • Page 14: “Towards body on chip models…. big data solutions”. Please elaborate or add appropriate references.

Reviewer 3 Report

This is a review paper for organ-on-a-chip industry toward broad adoption based on authors' survey of "end-users, experts and developers from both academia and industry" claimed by authors in this manuscript. My specific comments are as follows.

  1. This manuscript reports the statistic analyses of the survey for organ-on-a-chips' end-users, experts and developers from both academia and industry.  The survey list is attached as the supplementary.xlsx file.  To get convinced, the survey pool (who answer this survey) is strongly suggested to be released as a separate file.  The survey summary file, supplementary.xlsx, looks quite rough and not well organized. 
  2. The survey (or citation/study) of predicted market size for organ-on-a-chip categorized by different organ/disease is strongly suggested to be included/covered in this manuscript.     However, the market size information should be more comprehensive to make this manuscript valuable.  Also clearly define the reported market size shown in the revised manuscript.
  3. Without the well-defined survey pool list and the market size categorized into different organ-on-chip, this manuscript is just a statistic analysis of the survey data, which results from doubtful survey data. (The survey summary file, supplementary.xlsx, is also not well-done.)
  4. There are quite a lot of grammar mistakes and typos. Moderate English changes are required.

Author Response

  1. “This manuscript reports the statistic analyses of the survey for organ-on-a-chips' end-users, experts and developers from both academia and industry.  The survey list is attached as the supplementary.xlsx file.  To get convinced, the survey pool (who answer this survey) is strongly suggested to be released as a separate file.  The survey summary file, supplementary.xlsx, looks quite rough and not well organized.” 

We thank the Reviewer for highlighting this. We reorganized the excel file, including a more readable list of the questions, with the raw data for each answer, the normalized scores and two separate sheets with the list of originally contacted companies (names of contacts have been obscured for privacy constraints) and the details of the respondents that provided their contact details (sample pool).

  1. “The survey (or citation/study) of predicted market size for organ-on-a-chip categorized by different organ/disease is strongly suggested to be included/covered in this manuscript.     However, the market size information should be more comprehensive to make this manuscript valuable.  Also clearly define the reported market size shown in the revised manuscript.”

We have included in the new version a paragraph (lines 50 -54) with details on the market size and expected growth and two references to the most recent market research reports. In order to keep the focus of the research on the translational challenges, we decided not to include further details on the segmented markets, which are already covered by several publications.

  1. “Without the well-defined survey pool list and the market size categorized into different organ-on-chip, this manuscript is just a statistic analysis of the survey data, which results from doubtful survey data. (The survey summary file, supplementary.xlsx, is also not well-done.)”

As stated above, we have now included a new, reorganized excel accessible for further analysis which provides more details about the methods and channels used to collect data.

The market size relative to different organs is a topic well covered in published reports and other reviews. We thus included in the new version two references to those and a brief list of the most developed organs.

  1. “There are quite a lot of grammar mistakes and typos. Moderate English changes are required.”

We revised the manuscript and checked for all these mistakes. We hope this step, together with the proof reading phase, will help refine the manuscript.

Round 2

Reviewer 2 Report

The authors have answered my questions. Thank you for including the changes. 

Reviewer 3 Report

The revision looks ok to me.